# Alzheimer’s, Parkinson’s Disease and Amyotrophic Lateral Sclerosis Gene Expression Patterns Divergence Reveals Different Grade of RNA Metabolism Involvement

**DOI:** 10.3390/ijms21249500

**Published:** 2020-12-14

**Authors:** Maria Garofalo, Cecilia Pandini, Matteo Bordoni, Orietta Pansarasa, Federica Rey, Alfredo Costa, Brigida Minafra, Luca Diamanti, Susanna Zucca, Stephana Carelli, Cristina Cereda, Stella Gagliardi

**Affiliations:** 1Genomic and Post-Genomic Unit, IRCCS Mondino Foundation, 27100 Pavia, Italy; maria.garofalo@mondino.it (M.G.); cecilia.pandini@mondino.it (C.P.); orietta.pansarasa@mondino.it (O.P.); szucca@engenome.com (S.Z.); stella.gagliardi@mondino.it (S.G.); 2Department of Biology and Biotechnology “L. Spallanzani”, University of Pavia, 27100 Pavia, Italy; 3Dipartimento di Scienze Farmacologiche e Biomolecolari (DiSFeB), Centro di Eccellenza sulle Malattie Neurodegenerative, Università degli Studi di Milano, Via Balzaretti 9, 20133 Milano, Italy; matteo.bordoni@unimi.it; 4Department of Biomedical and Clinical Sciences “Luigi Sacco”, University of Milan, Via G.B Grassi 74, 20157 Milan, Italy; federica.rey@unimi.it (F.R.); stephana.carelli@unimi.it (S.C.); 5Pediatric Clinical Research Center Fondazione “Romeo ed Enrica Invernizzi”, University of Milano, Via G.B. Grassi 74, 20157 Milano, Italy; 6Unit of Behavioral Neurology, IRCCS Mondino Foundation, 27100 Pavia, Italy; alfredo.costa@mondino.it; 7Department of Brain and Behavioral Sciences, University of Pavia, 27100 Pavia, Italy; 8Parkinson Unit and Movement disorders Mondino Foundation IRCCS, 27100 Pavia, Italy; brigida.minafra@mondino.it; 9Neuro-Oncology Unit, IRCCS Mondino Foundation, 27100 Pavia, Italy; luca.diamanti@mondino.it; 10enGenomesrl, 27100 Pavia, Italy

**Keywords:** Alzheimer’s disease, Parkinson’s disease, amyotrophic lateral sclerosis, gene expression, long non-coding RNAs

## Abstract

Alzheimer’s disease (AD), Parkinson’s disease (PD), and amyotrophic lateral sclerosis (ALS) are neurodegenerative disorders characterized by a progressive degeneration of the central or peripheral nervous systems. A central role of the RNA metabolism has emerged in these diseases, concerning mRNAs processing and non-coding RNAs biogenesis. We aimed to identify possible common grounds or differences in the dysregulated pathways of AD, PD, and ALS. To do so, we performed RNA-seq analysis to investigate the deregulation of both coding and long non-coding RNAs (lncRNAs) in ALS, AD, and PD patients and controls (CTRL) in peripheral blood mononuclear cells (PBMCs). A total of 293 differentially expressed (DE) lncRNAs and 87 mRNAs were found in ALS patients. In AD patients a total of 23 DE genes emerged, 19 protein coding genes and four lncRNAs. Through Kyoto Encyclopedia of Genes and Genomes (KEGG) and Gene Ontology (GO) analyses, we found common affected pathways and biological processes in ALS and AD. In PD patients only five genes were found to be DE. Our data brought to light the importance of lncRNAs and mRNAs regulation in three principal neurodegenerative disorders, offering starting points for new investigations on deregulated pathogenic mechanisms.

## 1. Introduction

Aging-related neurodegenerative diseases, such as Alzheimer’s disease (AD), Parkinson’s disease (PD), and amyotrophic lateral sclerosis (ALS), among others, are characterized by the progressive degeneration of the structure and function of the central or peripheral nervous systems [1]. The onset of these diseases is growing as a consequence of the increased aging population [2] and it is thus necessary to fully decipher the molecular bases underlining these pathological conditions [3]. The progressive loss of motor neurons (MNs) is the primary feature of ALS, however its etiology remains unclear. ALS can be either familial (~10% of cases) or sporadic, and MN degeneration may be caused by common pathways leading to identical clinical appearance. Among the pathological hallmarks of this disease, of great relevance is the nuclear deficiency of RNA-binding proteins (RBPs) such as TDP-43 and FUS [4]. Amyloid plaques and neurofibrillary tangles (NFTs) are the main features of AD, along with other molecular alterations concerning neuropil threading, dystrophic neurites, neuroinflammation, and cerebral amyloid angiopathy [5]. The misfunctioning of these processes leads to neurodegeneration and thus macroscopic atrophy of the CNS. PD is a neurodegenerative disease characterized by the degeneration of dopaminergic neurons in the substantia nigra pars compacta (SNpc), which ultimately leads to impaired movements control [6].

Several common features of neurodegeneration have been studied and associated to the molecular pathogenesis of AD, PD, and ALS, and these include neuroinflammation, autophagy, and oxidative stress [7,8]. Moreover, pathogenic mutations, amyloid precursor proteins, superoxide dismutase, and DNA and RNA binding proteins are important contributing factors to these neurodegenerative diseases [9]. Although many common features have been found and studied, other emergent mechanisms remain unclear. The newly discovered role of RNA metabolism in these disorders is the object of recent studies [10] and still needs further investigation. Interestingly, recent evidences suggest that there are some common molecular events happening during the development and progression of these neurodegenerative disorders, which could be specifically driven by RNA perturbation [11]. Indeed, the disruption of RNA stability may play an important role in these pathologies. Alterations in RNA turnover have been identified in several disease pathways, including RNA sequestration in stress granules or foci, RNA transport, exosomes, alternative splicing, and retrotransposons [12]. As the importance of RNA metabolism is gaining more and more relevance, the role of both small and long noncoding RNAs has also been investigated. MicroRNAs directly interact with partially complementary target sites located in the 3′ UTR of target mRNAs and modulate their expression by repressing their translation into proteins [13]. Their involvement in the central nervous system (CNS) homeostasis, and their dysfunction in neurodegenerative disease are objects of intense study [14,15]. Lately, long noncoding RNAs were also found to be involved in neurodegenerative-related processes. For example, the antisense NEAT1_2 was reported to be upregulated in conjunction with increased paraspeckle formation in the spinal motor neurons of patients with ALS [16] and its overexpression in Huntington’s disease was described as a defensive mechanism against cell injury [17]. Its deregulation was also displayed in PD, where its function is controversial. In fact, NEAT1_2 is reported to exert a protective role both when up-regulated or down-regulated. In the first case, it is described as a provider of drug-inducible neuroprotection from oxidative stress [18], and when down-regulated it seems to suppress disease’s progression [19]. Indeed, lncRNAs are just now starting to be characterized, as can be seen by the number of conflicting studies concerning their role in disease pathogenesis.

When focusing on the molecular pathogenesis of neurodegenerative diseases, the role of RNA metabolism has been the object of recent investigations [20,21,22], but many RNA driven mechanisms remain unknown. Indeed, further studies are needed in order to unravel the role of aberrant gene expression in sporadic ALS (SALS), AD, and PD. There is also currently no evidence in the literature which focuses on analyzing the common or divergent deregulated transcriptional profile in these three disorders. Thanks to RNA sequencing (RNA-seq) a huge variety of RNA species are detectable, including mRNA, non-coding RNA, transcript isoforms and splice variants [23]. Through a meta-analysis and validation of gene expression in different groups of patients, the identification of disease-associated genes that may play a significant role in several diseases’ pathophysiology is possible.

In this paper, we report a whole transcriptome profiling of both long non-coding and coding RNAs in peripheral blood mononuclear cells (PBMCs) of unmutated SALS) patients [24], unmutated sporadic AD patients, unmutated sporadic PD patients, and matched controls. PBMCs were chosen as they have been reported to be source of protein and nucleic acid biomarkers in neurodegenerative disorders, also allowing for their use as cellular models for these diseases [24,25,26,27]. After transcriptome analysis, we focused on the identification of possible crossroads or deviations in the dysregulated pathways of AD, PD, and ALS. Our results pointed out a different degree of RNA metabolism and dysregulation involvement in AD, PD, and ALS, highlighting a different role for lncRNAs and mRNAs regulation in the most relevant neurodegenerative disorders, offering an interesting starting point for future investigations on the pathogenic mechanisms involved.

## 2. Results

### 2.1. RNA-Seq Differentially Expressed mRNAs and lncRNAs

Throughout the whole work, coding and non-coding genes were considered differentially expressed and retained for further analysis when they presented a |log2(disease sample/healthy control)| ≥ 1 and a FDR ≤ 0.1. The number of DE genes found was different in the three groups, as was their biotype. A summary of these results is represented in Table 1. The first evidence is relative to the number of DE transcripts among the different diseases. In fact, in SALS the total number of affected genes is 380, while in AD it is 25 and in PD only five. It is also important to highlight that, in both SALS and PD, the majority of DE genes belongs to non-protein coding class, but in AD only four transcripts out of 23 do not encode for proteins. By analyzing the DE genes lists (Appendix A), none were shared amongst the three conditions, but further bioinformatics and literature searches allowed us to identify some common features that will be highlighted in Section 2.3.2.

#### 2.1.1. Amyotrophic Lateral Sclerosis

In SALS patients, RNA-seq data reported 380 differentially expressed (DE) genes, 293 of which were lncRNAs (183 up-regulated and 110 down-regulated genes). Specifically, 184 out of 293 were reported as antisense, 81 out of 293 as lincRNAs, while the remaining 28 were classified as processed transcripts or intronic sense RNAs. Concerning coding genes, 87 differentially expressed mRNAs were identified, 30 of which were down-regulated whereas 57 were up-regulated. Heat-maps separately representing the expression levels of all dysregulated mRNAs and lncRNAs in SALS and healthy subjects are represented in Figure 1A,B. Different expression profiles in SALS and healthy controls can be visibly distinguished. The most deregulated coding genes were KIAA2013, HDAC1 and MYCBP. KIAA2013 is an uncharacterized protein and HDAC1 is an histone deacetylase whose aberrant expression has already been linked to ALS [28]. Interestingly, MYCBP is the binding protein of MYC, an important oncogene well characterized in cancer [29].

When considering the top 10 of DE lncRNAs, our data showed an interesting deregulation of antisense (AS) RNAs related to genes involved in transcription regulation pathways such as ZEB1-AS and ZBTB11-AS. Also, XXbac-BPG252P9.10 is described as the antisense transcript of IER3, involved in transcription. Interestingly, some of the sense genes regulated by the DE AS lncRNAs in SALS are already linked to neurodegenerative disease, such as UBXN7-AS [30], ATG10-AS38 [31], and ADORA2A-AS [32].

#### 2.1.2. Alzheimer’s Disease

In AD patients, a total of 23 DE genes has emerged, 19 of which were protein coding genes (eight up-regulated and 11 down-regulated) and four of which were non-coding genes, with three up-regulated lincRNAs and one down-regulated small nucleolar RNA. As the total number of DE in AD is limited, the heat-map (Figure 1C) shows the expression rate of both coding and non-coding genes in AD population versus matched healthy controls. Despite the low number of deregulated genes, there is a neat separation between the two groups, indicating a disease-specific gene expression pattern.

Amongst the most deregulated, we found coding genes already associated to AD such as TLN1 [33], and four nuclear pseudogenes (MTRNR2L6, MTRNR2L1, MTRNR2L10, and MTRNR2L8) of the mitochondrial MT-RNR2 gene [34]. VCL was also upregulated. This gene encodes for a protein involved in a pathway associated to Aβ toxicity [35] and associates to TLN1 [36].

When looking at the deregulated lncRNAs, three up-regulated lincRNAs were found: CH507-513H4.4, CH507-513H4.6, and CH507-513H4.3. These are novel transcripts, similar to YY1 associated myogenesis RNA 1 (YAM1), and they are reported as AD associated in LncRNADisease v2.0 Database [37].

#### 2.1.3. Parkinson’s Disease

In PD patients, only five genes were found to be DE, four of which were down-regulated (one protein coding, one small nucleolar RNA, one sense intronic transcript, and one lincRNA) and one of which was an up-regulated lincRNA. The heat-map reporting DE mRNAs and lncRNAs is shown in Figure 1D. The protein coding gene TBC1D3 has been associated to generation of basal neural progenitors [38]. The lncRNAs found in PD patients are currently not correlated to specific pathways.

### 2.2. Validation of Deregulated Coding and Non-Coding Genes

We selected specific mRNAs and lncRNAs for further qPCR validation, and two replicates per sample were performed. The validated DE RNAs were selected considering their fold change (FC), their previous description as “known”, antisense, and processed transcripts, and a balance between up- and down-regulated transcripts.

Transcripts validated in SALS are reported in Figure 2. Those validated in AD are reported in Figure 3 and PD validated ones are shown in Figure 4. Given the relevance of DE genes found in sALS patients, we decided to expand qPCR validation on a larger cohort of sporadic patients: 30 SALS and 30 healthy controls PBMCs underwent lncRNAs confirmation, while 10 SALS and 10 healthy controls PBMCs underwent mRNAs validation.

### 2.3. mRNA Pathway Analysis

#### 2.3.1. Amyotrophic Lateral Sclerosis

KEGG pathway analysis for the totality of DE mRNAs includes evidences of cancer-related pathways, longevity pathway and atherosclerosis (Figure 5). The GO biological processes enriched terms are related to intracellular transport, Reactive Oxygen Species (ROS) response, and regulation of transcription (Figure 6A).

Activating binding, transcription factor activity, NF-kappaB binding and activating transcription factor binding, and DNA binding were the most enriched GO terms in the Molecular Function database. (Figure 6B). Moreover, integral components of plasma membrane and chromatin resulted as enriched GO terms for cellular component (Figure 6C).

#### 2.3.2. Alzheimer’s Disease

KEGG pathways enriched by DE genes highlighted terms related to ribosome, focal adhesion, and atherosclerosis (Figure 7). The GO enriched terms for biological process are represented by exocytosis and protein targeting membrane (Figure 8A). GO molecular function analysis resulted in ribosome and focal adhesion (Figure 8B). With respect to cellular components, the most enriched GO concern phosphorylase and oxidase activity (Figure 8C).

KEGG pathway and GO terms enrichment analysis for DEs in SALS and AD patients compared to healthy controls has been performed for DE genes [39]. We did not conduce this analysis for PD because of the low number of resulting DE. Despite the fact that no common DE genes were found in the three cohorts of patients (Figure 9), some shared features related to KEGG and GO were reported (Table 2).

## 3. Discussion

Transcriptome analysis enables one to envision the complete biological context of disease pathogenesis by studying the gene expression profile. Moreover, the emergent role of various classes of regulatory non-coding RNAs (ncRNAs) in the onset of several diseases, such as cancer and neurodegenerative disorders, is becoming increasingly recognized. We performed a full profiling, via an RNA-Seq approach, of the lncRNAs and mRNAs in human PBMCs from SALS patients, AD patients, PD patients, and healthy controls in order to understand the coding and ncRNAs driven epitranscriptomics effect on the pathogenesis of these diseases.

Our work consisted in a deep screening of both coding and non-coding RNAs in ALS, AD and PD patients with the aim of describing the transcriptome alterations present in these diseases and assessing whether a shared deregulation was present in coding/noncoding RNAs.

Even if we did not find any common DE genes in the three cohorts of patients, some shared feature related to KEGG and GO terms were reported in SALS and AD, but not in PD as the number of DE genes made impossible the realization of this analysis. KEGG analysis revealed “dilated cardiomyopathy”, “complement and coagulation cascade” and “fluid shear stress” as deregulated pathways in both diseases. Interestingly, the specific genes involved in these pathways are different, and moreover it is possible to nota that in SALS the genes are both up-regulated and down-regulated, whereas in AD these pathways pertain only up-regulated genes. A similar situation is also found in Gene Ontology analysis, where we found an enrichment of “cellular response to reactive oxygen species” and “protein localization to cell surface”. In SALS group, the response to ROS is again represented by two up-regulated genes and one down-regulation of a gene.

Even if it was not possible to find overlapping terms between PD and the other two diseases, it is interesting to note that, amongst the KEGG pathways related to AD downregulated DE RNAs, the term “Parkinson’s disease” is present. Indeed, NDUFV2, down-regulated in AD patients, has been addressed as a genetic variation promoting a mild form of Parkinsonism with a prognosis similar to that of idiopathic PD [40].

The common pathways affected in both ALS and AD that emerged from KEGG analysis are extremely relevant in neurodegeneration pathophysiology. Indeed, “dilated cardiomyopathy” has already been associated to heart failure in both AD, because of Aβ amyloid accumulation in the heart [41], and ALS, because of sympathetic hyperactivity that leads to cardiac death [42], which is the second cause of death in these patients. Also, “complement and coagulation cascades” resulted from DE genes in both groups of patients. These two cascades are known to be regulated by a tight balance between protective and toxic effects and under molecular stress typical of neurodegeneration the signaling of these processes might be compromised promoting cellular damage [43]. It is also relevant to mention that one of the resulting cellular processes in GO analysis was “cellular response to reactive oxygen species”. This is highly relevant since the oxidative stress is one of the most studied molecular features of neurodegenerative diseases [44,45].

The crucial result that emerged from our RNA-seq analysis is the one concerning the small number of DE genes found in PD PBMCs. Despite the fact that an important deregulation of noncoding genes in CNS is observed [46], it has already been reported that, in peripheral blood, these RNAs are present but do not change their expression when compared to healthy controls [47,48]. In this group, we found the TBC1D3 gene to be downregulated. This transcript is not related to any PD molecular alteration, but it has been addressed as a promoter of the generation of basal neural progenitors and inductor of cortical folding in mice [38]. The only up-regulated gene was SCARNA2 (small Cajal body-specific RNA 2), a lincRNA which is independently transcribed [49]. Two other ncRNAs were found down-regulated, namely RP1-29C18.9 and RP1-29C18.8, but there is nothing reported in literature concerning their potential implication in PD.

Our data highlighted the different grade of RNA metabolism involvement in Alzheimer’s disease, amyotrophic lateral sclerosis, and Parkinson’s disease. Specifically, we found a great difference in the amount of differentially expressed genes in the three conditions, meaning that a milder transcriptional machinery involvement is present in AD and PD compared to ALS. Despite this imbalance, some molecular features common amongst them are present.

Considering our results, it emerged that a more extensive study of coding and noncoding genes modulation is needed in order to exploit novel mechanisms involved in the molecular pathogenesis of ALS, with the aim of identifying possible therapeutic targets.

## 4. Materials and Methods

### 4.1. Study Subjects

Briefly, 10 SALS, 6 AD, and 6 PD patients and 14 age- and sex-matched healthy controls (CTR) were recruited after obtaining their written informed consent (Table 3). All the subjects were deep-sequenced and included in Real Time PCR experiments. ALS, AD and PD patients underwent clinical and neurologic examination at IRCCS Mondino Foundation (Pavia, Italy). ALS patients were diagnosed following the El Escorial criteria [50]. AD diagnosis was made according to the National Institute of Neurological and Communicative Disorders and Stroke and the AD and Related Disorders Association (NINCDS-ADRDA) criteria [51]. For PD, patients Movement Disorder Society (MDS) clinical diagnostic criteria were used [52]. All patients were analyzed to exclude any mutations in causative genes by whole exome sequencing. The control subjects were recruited at the Transfusional Service and Centre of Transplantation Immunology, Foundation San Matteo, IRCCS (Pavia, Italy). The study protocol to obtain PBMCs from patients and controls was approved by the Ethical Committee of the IRCCS Mondino Foundation (Pavia, Italy), (Codes 20200045392; n°20170001758; n°2020042334). Before being enrolled, the subjects participating in the study signed an informed consent form (Code 375/04—version 07/01/2004). All experiments were performed in accordance with relevant guidelines and regulations.

### 4.2. Isolation of Human Peripheral Blood Mononuclear Cells and RNA Extraction Sequencing

PBMCs were prepared by centrifugation layering peripheral blood on Ficoll-Histopaque (density = 1.077) and centrifuged at 950 g for 30 min. After isolation, cell viability was assessed by a trypan blue exclusion test. Total RNA was isolated by Trizol reagent (Life Science Technologies, Monza, Italy) according to manufacturer’s instructions. Quantification and quality control of RNAs were performed using a Nanodrop ND-100 Spectrophotometer (Nanodrop Technologies, Wilmington, USA) and a 2100 Bioanalyzer (Agilent RNA 6000 Nano Kit, Waldbronn, Germany); RNAs with a 260:280 ratio of ≥1.5 and an RNA integrity number of ≥8 were selected for deep sequencing.

### 4.3. Libraries Preparation for RNA-Seq and Bioinformatic Data Analysis

Sequencing libraries of SALS and AD patients and matched controls were prepared with the Illumina TruSeq Stranded RNA Library Prep kit, version 2, Protocol D, using 500 ng total RNA (Illumina, San Diego, USA). For PD patients and controls libraries it has been used the SENSE Total RNA-Seq Library Prep Kit (Lexogen, Vienna, Austria), starting from 200 ng total RNA. Quality of sequencing libraries was assessed by 2100 Bioanalyzer with a DNA1000 assay and DNA High Sensitivity assay (Agilent, Waldbronn, Germany).

RNA processing was carried out using Illumina NextSeq 500 Sequencing (Illumina, San Diego, USA). FastQ files obtained from raw sequencing reads produced by Illumina NextSeq sequencer were generated via llumina bcl2 fastq2(Version2.17.1.14-http://support.illumina.com/downloads/bcl-2fastq-conversion-software-v217.htm). STAR/RSEM software was used for computing gene and transcript intensities [53] using Gencode Release 19 (GRCh37.p13) as a reference, using the “stranded” option. Number of transcripts having at least 10 counts in the three diseases are reported in Appendix A. R package EBSeq was used for differential expression analysis of mRNA was performed using [54], because of its higher performance in identifying isoforms differential expression [55], while for long non-coding RNAs the R package DESeq.2 was employed [56]. Coding and non-coding genes were considered differentially expressed and retained for further analysis with |log2(disease sample/healthy control)| ≥ 1 and a FDR ≤ 0.1. This choice is motivated by the decision to maximize the sensitivity of this analysis, in order to perform a massive screening. RNA sequencing data are available in GEO repository (GSE106443; GSE161199).

### 4.4. Pathway Analysis

Coding genes underwent gene enrichment analysis [39]. To do so, we used Gene Ontology 2018 (GO) for biological processes, cellular components, and molecular function and KEGG 2019 (Kyoto Encyclopedia of Genes and Genomes http://www.genome.ad.jp/kegg) for pathway analysis via the enrichR web tool [57,58].

### 4.5. Real Time PCR

Primer 3.0 was used to design PCR oligonucleotides specifically targeting and amplifying mRNA, avoiding non-specific amplification products. Moreover, primers were chosen so as to not overlap with antisense sequences (primers upon request). An iScript™ cDNA Synthesis Kit (BioRad, Richmond, CA, USA) was used for preparing total cDNAs. Real time PCR (qPCR) reactions were performed with SYBR Green SuperMix (BioRad, Richmond, CA, USA), using 1 μL of cDNA template (or water control). Cycle threshold (Ct) values were normalized against those determined for GAPDH. Determination of fold-expression differences relative to healthy controls were carried out using the 2ΔΔCt method. Student’s t-test was used for the determination of significance of gene expression changes relative to controls using Prism GraphPad 5.02 software (GraphPad Software, San Diego, CA, USA).

## Figures and Tables

**Figure 1 ijms-21-09500-f001:**
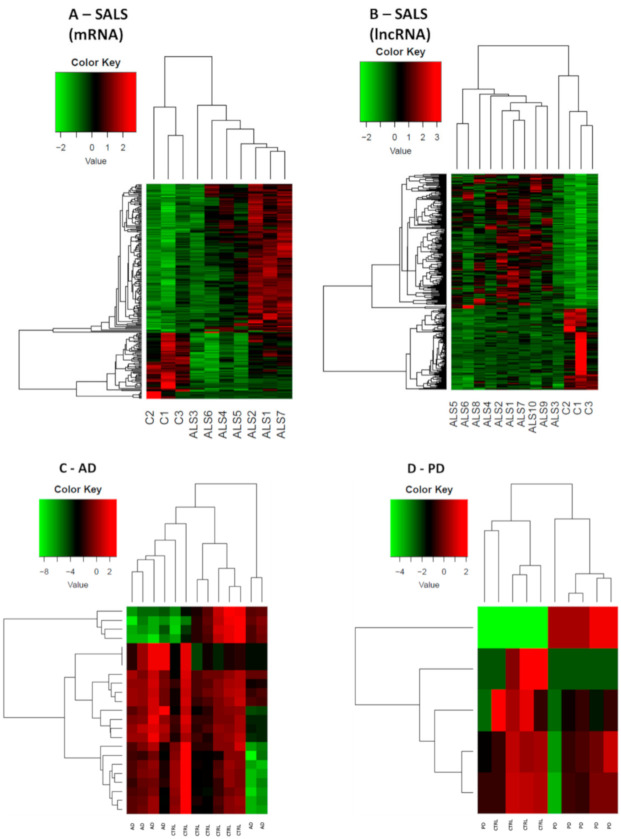
Expression profiles of differently expressed genes in SALS, AD, PD and healthy controls. In panel (**A**), SALS differentially expressed mRNAs are shown, while in panel (**B**) SALS differentially expressed lncRNAs are shown. 10 sALS and 3 CTRLs were used for this analysis. In panel (**C**) and (**D**) both mRNAs and lncRNAs DE respectively in AD and PD are shown, given the lower amount of DE genes. For both AD and PD heatmaps, 6 patients’ samples and 6 CTRLs were used. All comparisons are given between the disease state and the control samples. We considered as differentially expressed only genes showing |log2(disease sample/healthy donor)| ≥ 1 and a False Discovery Rate ≤ 0.1.

**Figure 2 ijms-21-09500-f002:**
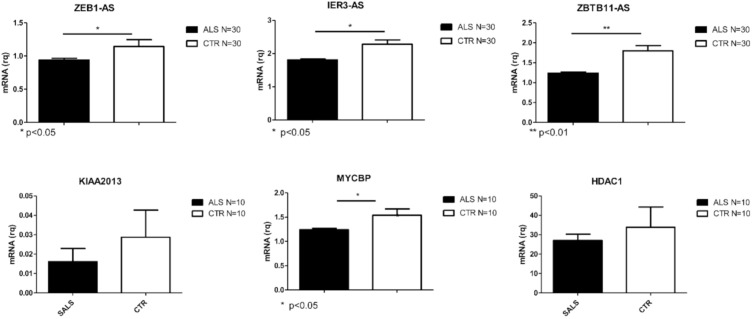
Differentially expressed transcripts verified by Real Time PCR in PBMCs from a larger cohort of SALS and CTRLs (N = 30 for lncRNAs and N = 10 for coding RNAs. * *p* < 0.05, ** *p* < 0.001 vs. CTR).

**Figure 3 ijms-21-09500-f003:**
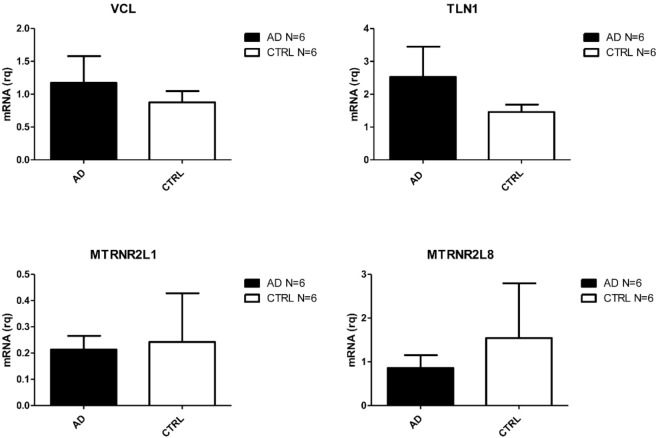
Differentially expressed transcripts verified by Real Time PCR in PBMC from AD and CTRs (N = 6).

**Figure 4 ijms-21-09500-f004:**
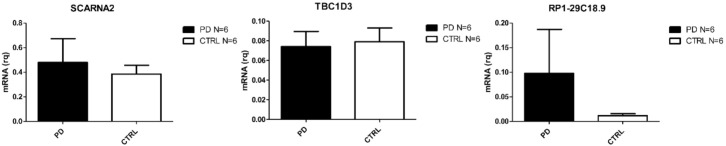
Differentially expressed transcripts verified by Real Time PCR in PBMC from PD and CTRs (N = 6).

**Figure 5 ijms-21-09500-f005:**
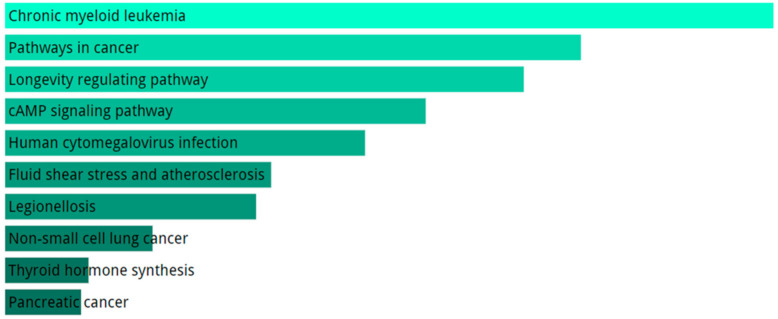
Analysis of pathways performed with KEGG using DE genes in SALS group compared to healthy controls. We only show top 10 KEGG terms. The significance of the specific gene-set term is represented by the length of the bar. The significance of the term is indicated by the brightness of the bar’s color (the brighter, the more significant).

**Figure 6 ijms-21-09500-f006:**
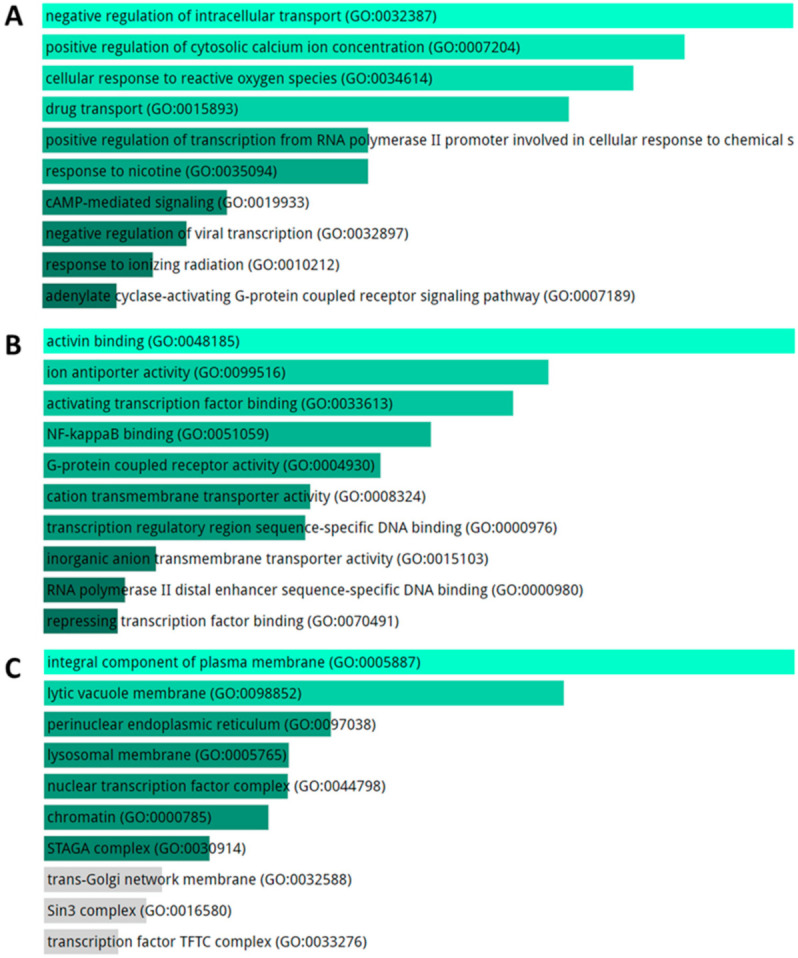
GO analysis for DE genes in SALS patients compared to healthy controls. TOP10 enriched GO terms for biological process (**A**), molecular function (**B**) and cellular component (**C**) The length of the bar represents the significance of that specific gene-set or term. The brighter the color, the more significant that term is, and the grey color refers to non-significant terms.

**Figure 7 ijms-21-09500-f007:**
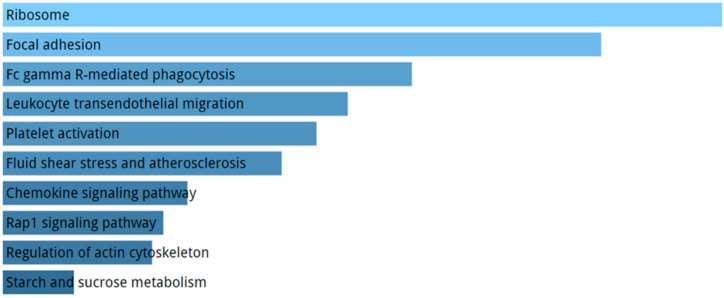
Pathways analysis performed with KEGG using DE genes in AD group compared to healthy controls. We report the top 10 KEGG terms. The significance of the specific gene-set term is represented by the length of the bar. The significance of the term is indicated by the brightness of the bar’s color (the brighter, the more significant).

**Figure 8 ijms-21-09500-f008:**
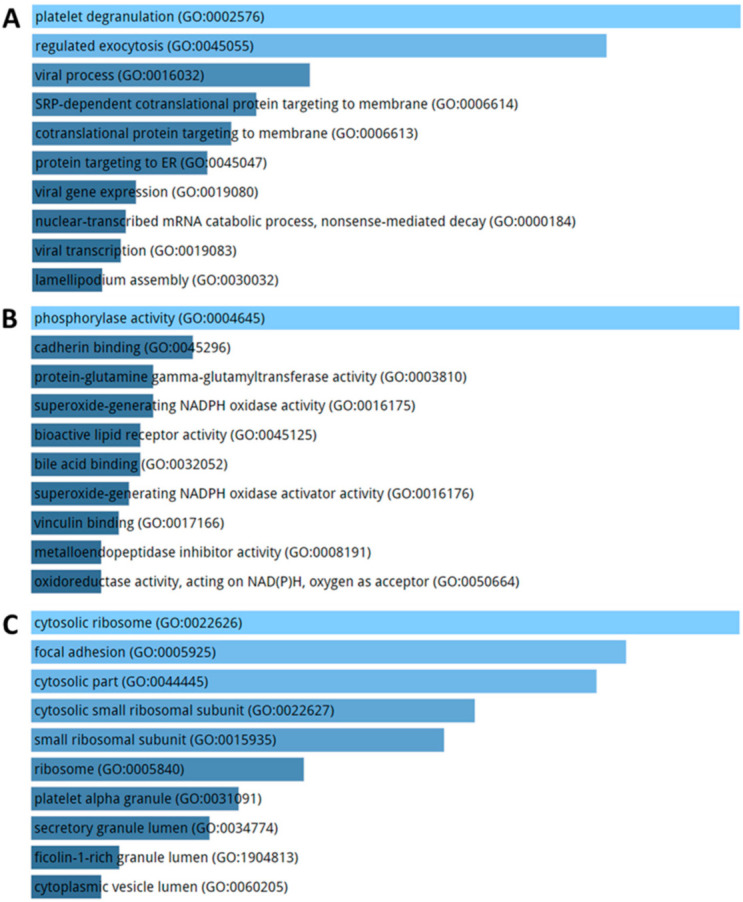
Biological process (**A**), molecular function (**B**) and cellular component (**C**) enrichment analysis performed on DE genes in AD patients compared to healthy controls. The significance of the specific gene-set term is represented by the length of the bar. The significance of the term is indicated by the brightness of the bar’s color (the brighter, the more significant).

**Figure 9 ijms-21-09500-f009:**
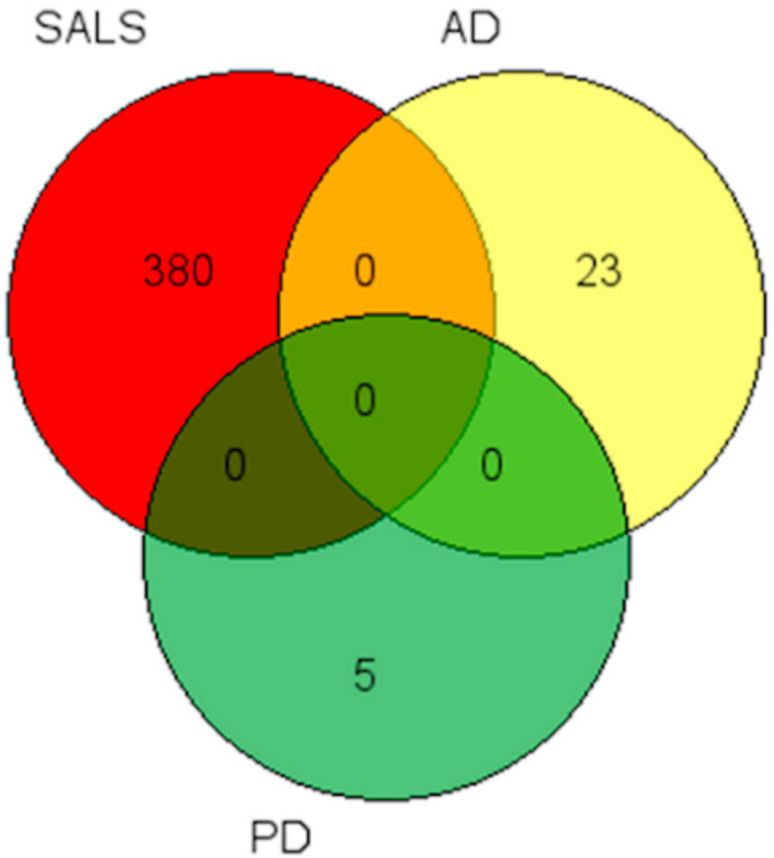
Venn diagram shows differentially expressed genes in common among the three conditions. No common deregulated gene emerged from DE analysis. In red the number of DE genes in SALS, in yellow the number of DE genes on AD and in green the number of DE genes in PD.

**Table 1 ijms-21-09500-t001:** Number of statistically significant differentially expressed mRNAs and lncRNAs in PBMCs from SALS, AD and PD patients, in terms of up-regulated transcripts, down-regulated transcripts and total.

	SALS		AD		PD	
	mRNA	lncRNA	mRNA	lncRNA	mRNA	lncRNA
Up-regulated	57	183	8	3	0	1
Down-regulated	30	110	11	1	1	3
Total	87	293	19	4	1	4

**Table 2 ijms-21-09500-t002:** Shared enriched KEGG or GO terms among SALS and AD patients. In the first column the term is reported; in the columns relative to each disease the genes with the highest fold change involved in each term are reported, in green those up-regulated, in red those down-regulated.

KEGG/GO Term	SALS	AD
Dilated cardiomyopathy (DCM)	ADCY9 PLN	ITGA2B
Complement and coagulation cascades	A2M F2RL2	F13A1
Fluid shear stress and atherosclerosis	RELA ACVR2A NFE2L2	NCF1 ITGA2B
Cellular response to reactive oxygen species	MPV17L RELA NFE2L2	NCF1
Protein localization cell surface	SMURF1	VCL

**Table 3 ijms-21-09500-t003:** Baseline characteristics of subjects recruited for this study.

	SALS	AD	PD	CTRL
	n = 10	n = 6	n = 6	n = 14
Age (M ± SD)	66.6 ± 10.1	78.2 ± 8.0	62.7 ± 6.7	53.2 ± 8.3
Sex				
Males n (%)	45%	34%	83%	64%
Females n (%)	55%	66%	17%	32%

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
