# Peer review of "Alzheimer’s, Parkinson’s Disease and Amyotrophic Lateral Sclerosis Gene Expression Patterns Divergence Reveals Different Grade of RNA Metabolism Involvement"

_ijms, 2020, doi:10.3390/ijms21249500_

Round 1

Reviewer 1 Report

This study looks at coding and non-coding RNA expression in peripheral blood mononuclear cells (PBMCs) from ALS, AD and PD patients and age-matched controls with the goal of identifying common mechanisms underlying the three diseases. However, rather than supporting their hypothesis, the results do not strongly suggest that common mechanisms underlie the different diseases but rather that disease-specific changes occur that may have some downstream overlaps between diseases. Thus, the results are interesting. However, a number of changes and clarifications are needed to make this a useful and informative paper. These are listed below. In addition, the English is quite hard to understand with numerous grammatical and word use errors. Thorough editing by a native English speaker is needed.

  1. The authors need to justify the use of PBMCs for this study.
  2. The authors need to make clear at the beginning of the results how differentially expressed genes are defined.
  3. The numbers of subjects in the different groups varies throughout the manuscript. Section 4.1 says 6 each of AD, PD and ALS and 13 controls but Table 3 says 10 ALS patients and 14 age matched controls while Figure 2 shows very different numbers of ALS patients and controls (10 and 30). The number of samples analyzed absolutely needs to be clarified. If it is not the same for each disease, then that could make a big contribution to the differences in the results with regard to DE genes because this might not be as obvious with smaller numbers of samples as appears to be the case for the AD and PD patients. This concern is reinforced by the heat maps which show quite a lot of variability among the AD and PD patients.
  4. All of the samples need to be shown in the heat maps. The authors also need to clarify how many samples were used for the DE gene analysis for each disease and the controls.
  5. The authors need to at least discuss the large differences in sex and/or age between the controls as compared with the AD and PD patients and how that might impact their results.
  6. The Discussion should incorporate previous studies using mRNA seq to look at differences between these disease populations. How do previous studies compare to the ones in this manuscript? Have others used PBMCs?
  7. Lines 244-247 make no sense. The sentence starts out comparing results for AD and PD patients but then shifts to AD and ALS patients. What is being compared here?

Reviewer 2 Report

The underlying concept behind this study is sound but currently the manuscript has not fully investigated the proposal. Some issues are not discussed or expanded upon. Currently the PD is too limited for any meaningful conclusions to be drawn. It is interesting that this group was prepared for sequencing using a different approach - is this having a n impact upon the data? Admittedly commercial kits are used in each case but do they differ in terms of the numbers of transcripts identified prior to the differential expression analysis? It is quite possible that only highly expressed transcript are being examined.There is a discrepancy between the text and the Table 3 in terms of ALS patient numbers. I wonder if the stringency of the analysis has eliminated some possible interesting transcript changes. The heat maps of the differential expression changes seem to indicate the existence of subgroups in the data but this is not discussed or further investigated. In that sporadic cases are being examined there may be underlying patterns of difference that are important. Some of the relative concentrations determined by the RT-qPCR are extreme indicating either extremely high or low concentrations of the gene of interest relative to the comparator. It is not clear how many replicates were carried out to confirm these data. The KEGG and GO analyses are hampered for all of the analyses, not just the PD, as a result of the small numbers of differentially expressed transcripts identified. This makes strongly supported conclusions difficult. 

Overall I feel that using two different kits has hampered some of the experimental work and the analysis has failed to identify subtle changes which could be of interest. 

Round 2

Reviewer 1 Report

The authors have adequately addressed my concerns.

Reviewer 2 Report

Thank you for addressing my questions and resolving the issues I had,